# Treatment burden in survivors of prostate and colorectal cancers: a qualitative interview study

Rosalind Adam ,[1] Lisa Duncan ,[1] Sara J Maclennan,[2] Louise Locock[3]

¹Academic Primary Care, Institute of Applied Health Sciences, University of Aberdeen, Aberdeen, UK
²Academic Urology Unit, Institute of Applied Health Sciences, University of Aberdeen, Aberdeen, UK
³Health Services Research Unit, Institute of Applied Health Sciences, University of Aberdeen, Aberdeen, UK

**Correspondence to**
Dr Rosalind Adam;
rosalindadam@abdn.ac.uk

## ABSTRACT

**Objectives** Treatment burden is the workload of healthcare and the impact this has on the individual. Treatment burden is associated with poorer patient outcomes in several chronic diseases. Illness burden has been extensively studied in cancer, but little is known about treatment burden, particularly in those who have completed primary treatment for cancer. The aim of this study was to investigate treatment burden in survivors of prostate and colorectal cancers and their caregivers.

**Design** Semistructured interview study. Interviews were analysed using Framework and thematic analysis.

**Setting** Participants were recruited via general practices in Northeast Scotland.

**Participants** Eligible participants were individuals who had been diagnosed with colorectal or prostate cancer without distant metastases within the previous 5 years and their caregivers. Thirty-five patients and six caregivers participated: 22 patients had prostate and 13 had colorectal cancers (six male, seven female).

**Results** The term 'burden' did not resonate with most survivors, who expressed gratitude that time invested in cancer care could translate into improved survival. Cancer management was time consuming, but workload reduced over time. Cancer was usually considered as a discrete episode. Individual, disease and health system factors protected against or increased treatment burden. Some factors, such as health service configuration, were potentially modifiable. Multimorbidity contributed most to treatment burden and influenced treatment decisions and engagement with follow-up. The presence of a caregiver protected against treatment burden, but caregivers also experienced burden.

**Conclusions** Intensive cancer treatment and follow-up regimens do not necessarily lead to perceived burden. A cancer diagnosis serves as a strong motivator to engage in health management, but a careful balance exists between positive perceptions and burden. Treatment burden could lead to poorer cancer outcomes by influencing engagement with and decisions about care. Clinicians should ask about treatment burden and its impact, particularly in those with multimorbidity.

**Trial registration number** NCT04163068.

## STRENGTHS AND LIMITATIONS OF THIS STUDY

⇒ In-depth interviews with 41 individuals, generating rich qualitative data.
⇒ Participant feedback on the results and key messages.
⇒ The study was conducted in a single geographical area.
⇒ Participants from areas of higher social deprivation were relatively under-represented.

There is a move away from hospital and clinician-led cancer care towards supported self-monitoring and self-management by patients and their families in the community.[5 6]

Treatment burden is the workload of healthcare and the impact that this work has on the individual.[7] Treatment burden is of increasing importance[8] due to ageing populations, a rising prevalence of multimorbidity[9 10] and increased pressure on healthcare systems. Healthcare workload can encompass a wide range of tasks, including 'sense-making' work,[11] monitoring/managing symptoms, managing medicines, navigating the healthcare system and changing health-related behaviours.[8 11]

Treatment burden and illness/disease burden are closely linked but conceptually different. Illness burden describes the impact of an illness on an individual, such as morbidity and mortality.[12] The actions taken to manage health and their consequences can lead to treatment burden.[13–15]

Sav *et al* noted six key domains of treatment burden, encompassing 'financial, medication, administrative, time/travel, lifestyle, and healthcare' dimensions, and 'antecedents' which can influence the severity of treatment burden, such as age, gender, treatment characteristics and disease type.[16 17] Having good social support or a caregiver can lower treatment burden for patients,[17] but caregivers can also become burdened.[18 19] The impact of treatment burden on informal caregivers is under-researched.[18]

## INTRODUCTION

Modern cancer survivorship care places physical, financial, psychosocial and practical demands on individuals and their families.[1–4]

Treatment burden is likely to be important in cancer survivors (The term 'survivor' is used here to describe individuals living with and beyond cancer. It is a term that is widely used in research, guidelines and charities but which can divide opinion in some individuals with cancer) and their caregivers. Questionnaire studies have detected high levels of treatment burden after cancer in patients, particularly in those with lower health literacy,[20 21] multiple comorbidities[20–23] and lower social support.[20 22] One qualitative interview study investigated treatment burden in individuals who were undergoing or had recently completed treatment for lung cancer.[24] Only one caregiver participated in the study. Patients had restructured their lives to accommodate treatment-associated workload. This was accepted by patients as a necessity due to the severe and life-threatening nature of lung cancer.[24] It is unclear how treatment burden might be perceived by survivors of different types of cancer in whom the prognosis is more favourable than lung cancer.

Prostate and colorectal cancers are the third and fourth most common invasive cancers worldwide.[25] They have excellent prognoses when detected and treated early.[3 26] Prostate and colorectal cancers were chosen to explore treatment burden in this study because they encompass a wide range of treatment modalities (such as surgery, radiotherapy, and chemotherapy), lasting sequelae (eg, fatigue, persistent pain, incontinence, sexual problems and stoma management) and follow-up activities. Individuals play a key role in improving their own prognosis by self-monitoring for symptoms and attending for scans and blood tests to detect recurrence, and by adhering to diet and exercise recommendations,[4 27] and may therefore be at risk of treatment burden. Informal caregivers are key supporters of these activities.[28]

The aim of this study was to investigate perceptions of treatment burden in individuals who had completed treatment for colorectal or prostate cancer and their caregivers and the impact of treatment burden on these individuals. It is held that individuals who become over-burdened by the workload of healthcare disengage from self-management activities, leading to poorer outcomes.[21–23] Treatment burden could be an important mediator of poorer outcomes in cancer survivors and patients/caregivers are best placed to give insights into mechanisms of treatment burden and aspects that are potentially modifiable.

## METHODS
### Setting and design
A qualitative semistructured interview study was conducted in National Health Service (NHS) Grampian in Northeast Scotland. The NHS is a publicly funded healthcare system which is free at the point of delivery. In Grampian, cancer care is centred around a university teaching hospital in Aberdeen with academic links and care pathways that are integrated with local cancer charities[29]. Grampian had a 2011 census population of 569 061, and around one-third of the population live rurally.[30]

### Participants and recruitment
Eligible participants were adults with a history of localised or locally advanced prostate or colorectal cancer, diagnosed within the past 5 years. A 5-year cut-off was chosen so that individuals were reflecting on recent experiences of cancer treatment and follow-up, and because many individuals with low-risk disease are discharged from follow-up after 5 years. Participants were included if they had received any cancer treatment/management, including, and not limited to, active surveillance, surgery, radiotherapy or chemotherapy. Individuals who were undergoing or on waiting lists for chemotherapy, radiotherapy or surgery were excluded because their experiences of treatment burden were likely to be different from individuals who were in the follow-up stages after active cancer treatment. Those with distant metastases were excluded because treatment and follow-up for these individuals would have different aims and formats of delivery.

NHS Research Scotland Primary Care (NRS Primary Care) assisted with recruitment. NRS Primary Care recruited general practices and searched electronic medical records using Read codes for prostate and colorectal cancers. General practices sent invitation packs to eligible patients, and those interested in participating responded directly to the research team. Eligible patients were invited to nominate a caregiver to participate in the interview. Separate invitation letters and information sheets were included for caregivers in packs sent to patients.

### Data collection and management
An interview topic guide (online supplemental file 1) was designed to ensure comprehensive coverage of the topic. A conceptual model (online supplemental figure 1) was produced, drawing on existing literature on treatment burden in stroke,[11] normalisation process theory (NPT)[31] and the health action process approach (HAPA).[32] NPT has been used to understand how individuals embed new practices within their daily life and has been a useful model through which to explore treatment burden after stroke.[11] The HAPA describes both how individuals become motivated to engage with healthcare or self-management work, and how individuals then translate this motivation into engagement with and maintenance of self-management practices over time. The HAPA integrates and extends previous behavioural theories by including a range of important constructs such as self-efficacy and intention, which can predict and explain human behaviours.[33]

The conceptual model was used to inform the topic guide, which included questions about the initial diagnosis and treatment period, current self-management strategies, social support and interactions with others. Participants were asked to reflect on the work of cancer

management and any effects that managing their health had on them. The guide was used flexibly, allowing the interview to be directed by participant responses. Participants and caregivers were asked about the same topics and interviewers adapted the questions during the interview to ensure that caregiver perceptions were fully captured.

The study commenced in January 2020 and was put on hold in March 2020 during the COVID-19 pandemic, restarting in May 2021. Interviews conducted prior to March 2020 were conducted face to face or by telephone, according to participant preference. All interviews undertaken after May 2021 were conducted by telephone. Interviews were conducted by two authors (RA and LD), were audio recorded and transcribed verbatim by a university-approved professional transcription company. Transcripts were checked for accuracy, anonymised and then imported into NVivo software (QSR International. V.12, 2018) for analysis.

### Data analysis

Framework analysis[34] and thematic analysis[35] were used to organise and analyse the data. A framework matrix was created with headings. Notes and quotations from each interview were listed under headings to summarise study data so that all authors could have an overview of key data from the whole sample.

Thematic analysis was conducted on individual interview transcripts and involved: familiarisation; generating initial codes; generating initial themes; reviewing themes; defining and naming themes; and reporting the thematic analysis.[35] Familiarisation with data was achieved through checking, reading and re-reading of transcripts. Initial themes were generated by examining relationships between codes and common features in the data. Themes were revised and developed through discussions among authors. Data collection and analysis were conducted in tandem, and data collection continued until no new concepts or themes were found.

The authors' disciplinary backgrounds include general practice, health psychology and health services research.

### Patient and public involvement

Patients were not involved in the design of this qualitative study, but a key aim of this work was to understand patient and caregiver perceptions and opinions. Participants were sent a summary of the study results and invited to comment on them. Three participants replied and provided general updates on their progress. None suggested any changes or additions to the results.

### RESULTS

One hundred and sixty-three study invitations were sent by six general practices. Thirty-five patients and six caregivers (joint interviews) participated in 13 in-person face-to-face interviews and 22 telephone interviews (21% response rate). Participant-level demographics are presented in table 1. Participants were aged between 37 and 91 years, mean age 69 years. Twenty-two participants had prostate cancer and 13 had colorectal cancer (seven female, six male). Most participants were from areas of low socioeconomic deprivation, with 23/35 patients in deciles 6–10 of the Scottish Index of Multiple Deprivation[36] (where 1 is most deprived and 10 is least deprived). Most participants were from a large urban area (45.7%) or an accessible rural area (28.6%).

Comparing the demographics of those who participated to those who were invited but did not respond, a higher proportion of non-respondents resided in areas of higher socioeconomic deprivation (42.2% of non-respondents compared with 20% of participants). These differences were not statistically significant (see online supplemental table 1).

Interviews lasted between 21 and 94 min duration, mean 44 min.

### Thematic analysis

Themes and their interactions are summarised in figure 1.

#### Theme 1: getting on with the work of cancer survivorship

Participants described having little choice but to engage in the work of managing cancer and its sequelae because cancer was a threat to life. Several participants mentioned '*getting on with it*'.

The work of health management after completion of treatment involved planning, organisation, problem solving and psychological work. Individuals developed new routines, for example, pacing daily activities to manage fatigue, and preplanning before travelling (eg, packing necessary stoma equipment or medications).

> [The nurse] says to me, 'The difficult work will start after the operation', and I thought, 'What does she mean by that? The operation must be the difficult part'. But I now know by this what she means, (…) she says, 'you're going to be wearing pads and everything like that'. If I go to the town (…), a man doesn't seem to worry about where a toilet is. But now I need to find a toilet (…) but make sure it's got a bin. (Patient 7, male, aged 60–69 years, prostate cancer)

Participants took active approaches to psychological adjustment after cancer. Many made comparisons with others whom they felt were '*worse off*' (eg, more advanced cancer, other disabling chronic diseases). Some spoke of being '*determined*' or '*positive*'. For many, cancer was always present in the background, and work was required to return to '*near*' normality after treatment.

> I do feel that I totally lost two years of my life, because I think it took that time to get through the operation, get back to feeling normal (…) I think everything had changed about my life, in a way it had stopped and I've started a new life now (…) you think things will never be normal again, but you find a new normal. (Participant 15, female, aged 60–69 years, colorectal cancer)

**Table 1** Participant demographics

| Patient ID | Age range (years) | Sex | Cancer site | SIMD decile* | Urban-rural classification† | Comorbidities‡ (n) | Carer participated |
|---|---|---|---|---|---|---|---|
| 1 | 70–79 | M | Prostate | 10 | 1 | 0 | No |
| 2 | 80–89 | F | Colorectal | 8 | 1 | 1 | No |
| 3 | 90–99 | F | Colorectal | 7 | 1 | 2 | No |
| 4 | 60–69 | M | Prostate | 10 | 1 | 1 | No |
| 5 | 70–79 | M | Colorectal | 3 | 1 | 2 | No |
| 6 | 70–79 | M | Prostate | 5 | 5 | 2 | Yes |
| 7 | 60–69 | M | Prostate | 5 | 1 | 1 | No |
| 8 | 60–69 | M | Prostate | 7 | 5 | 1 | Yes |
| 9 | 70–79 | M | Prostate | 7 | 5 | 1 | Yes |
| 10 | 60–69 | M | Colorectal | 8 | 3 | 1 | No |
| 11 | 60–69 | M | Prostate | 7 | 5 | 0 | No |
| 12 | 60–69 | M | Colorectal | Missing | Missing | 1 | No |
| 13 | 60–69 | F | Colorectal | 9 | 5 | 0 | No |
| 14 | 70–79 | M | Prostate | 10 | 3 | 0 | No |
| 15 | 60–69 | F | Colorectal | 9 | 5 | 1 | Yes |
| 16 | 60–69 | M | Prostate | 3 | 1 | 4 | No |
| 17 | 60–69 | M | Colorectal | 7 | 1 | 1 | No |
| 18 | 60–69 | M | Prostate | 10 | 1 | 1 | No |
| 19 | 80–89 | M | Colorectal | 8 | 1 | 2 | Yes |
| 20 | 70–79 | M | Prostate | 2 | 1 | 1 | No |
| 21 | 70–79 | M | Prostate | 6 | 1 | 1 | No |
| 22 | 60–69 | M | Prostate | 9 | 6 | 4 | No |
| 23 | 50–59 | M | Prostate | 2 | 1 | 1 | No |
| 24 | 70–79 | M | Prostate | 9 | 5 | 1 | No |
| 25 | 70–79 | M | Prostate | 5 | 1 | 1 | No |
| 26 | 70–79 | M | Prostate | 10 | 1 | 5 | No |
| 27 | 70–79 | M | Prostate | 6 | 1 | 1 | No |
| 28 | 70–79 | M | Prostate | 8 | 6 | 3 | No |
| 29 | 50–59 | M | Prostate | 7 | 5 | 0 | No |
| 30 | 50–59 | F | Colorectal | 7 | 1 | 0 | No |
| 31 | 30–39 | M | Colorectal | 4 | 5 | 0 | No |
| 32 | 70–79 | F | Colorectal | 3 | 5 | 0 | No |
| 33 | 70–79 | F | Colorectal | 4 | 6 | 3 | No |
| 34 | 60–69 | M | Prostate | 5 | 6 | 0 | Yes |
| 35 | 60–69 | M | Prostate | 5 | 2 | 0 | No |

*Scottish Index of Multiple Deprivation (SIMD) is derived from postcode; 1 indicates most deprived areas and 10 least deprived.
†Sixfold urban-rural classification is also derived from Scottish postcode: 1=large urban areas, 2=other urban areas, 3=accessible small towns, 4=remote small towns, 5=accessible rural areas, 6=remote rural areas.
‡Comorbidities were self-reported by patients and were chronic conditions (not including prostate or colorectal cancer), requiring ongoing treatment or management. Other primary cancers were also included. Complications of cancer treatment (eg, peripheral neuropathy, urinary incontinence) were not included as separate comorbidities. Individual conditions have not been listed so as to maintain participant anonymity.
F, female; M, male.

Health behaviour change was a source of work for some participants, but mainly in the period immediately after cancer diagnosis. Examples included efforts to lose weight prior to prostate cancer surgery to improve urinary continence or increasing exercise levels to prehabilitate before colorectal cancer surgery.

Few participants described efforts to make dietary, exercise or other healthy behaviour changes after completion

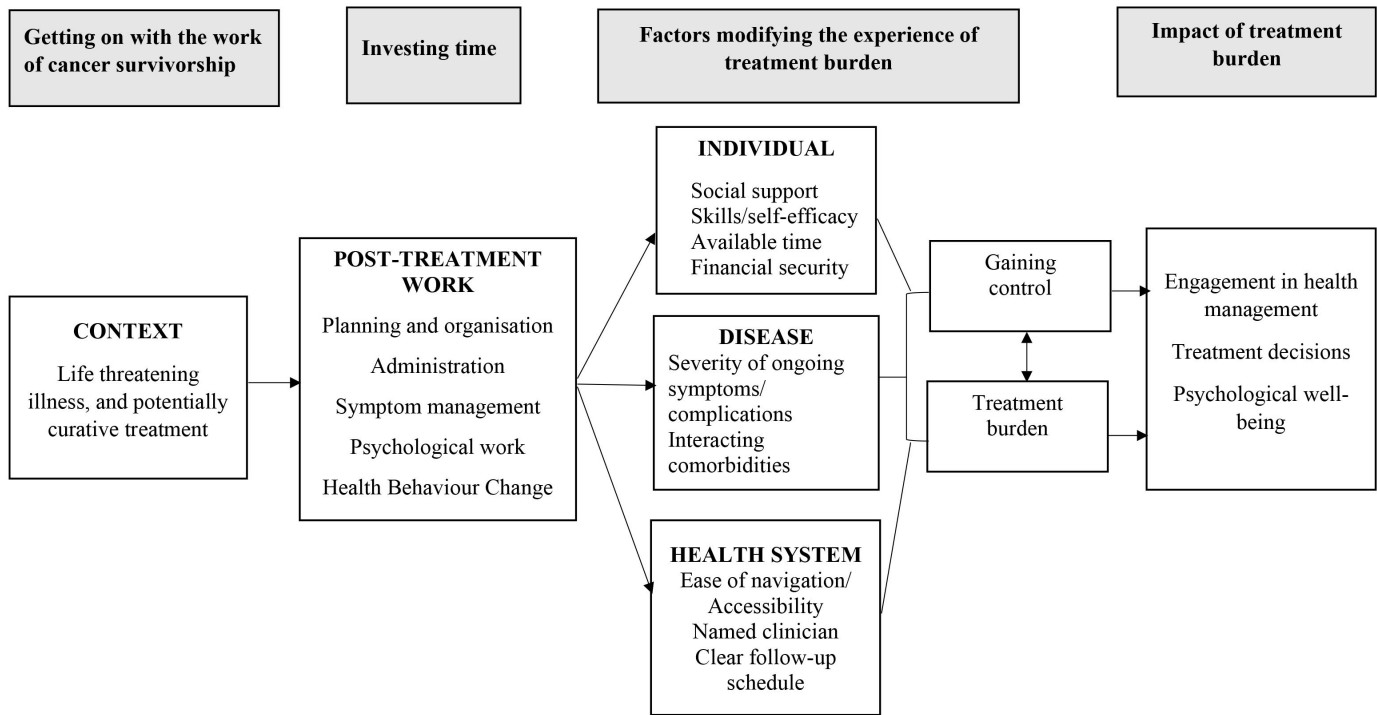

**Figure 1** Themes and their interactions.

of treatment. Some considered that they already lived healthy lifestyles, while others considered cancer to have been cured and no longer requiring attention. None of the participants spoke of specific efforts to reduce their risk of cancer recurrence. Participants with both prostate and colorectal cancers suggested that there was little emphasis placed on adopting healthy behaviours by healthcare professionals after their initial treatment had finished.

> It took a while for my wound to heal, I'm terribly overweight (…) I was bothered because I just didn't stop eating, which I think had nothing to do with the cancer. But I couldn't stop eating and I did mention it to the oncology nurse, and she said it was probably post-stress disorder and not to worry about it. (Patient 33, female, aged 70–79 years, colorectal cancer)

### Theme 2: investing time

Cancer management was time consuming during the initial treatment stages, but treatment-related work reduced over time, even when there were lasting effects or complications of treatment.

> Cancer management. Certainly, if you want to put it in terms of a graph, the graph has gone up steadily and quite steeply immediately post operation, but it has levelled out and is now on the way down and I'm pleased with that. (Patient 34, male, aged 60–69 years, prostate cancer)

Follow-up healthcare appointments and investigations to detect recurrence were viewed positively and were reassuring. Many spoke of their gratitude that cancer was detected at a potentially curable stage and that time invested in treatment and follow-up could translate into a greater amount of time to spend in the future on meaningful activities. Despite cancer management consuming considerable amounts of time, this was not perceived to be burdensome.

> I have a life and the treatment is actually preserving that very precious thing for me, so that means I'm going to hopefully be around for longer to be with my daughters and my grandchildren and I will be extended by this treatment (…) that to me is the most important thing. (Patient 7, male, aged 60–69 years, prostate cancer)

### Theme 3: factors modifying the experience of treatment burden

Participants identified individual-related, disease-related and health system-related factors that protected against or increased the likelihood of them experiencing treatment burden. At the level of the individual, the presence of a caregiver or a good social support network lessened the treatment burden associated with cancer. Spouses and other family members shared in the practical and psychological work of cancer management, from driving to and being present at appointments and undertaking domestic tasks, to giving input into treatment decisions and providing emotional support. During interviews, caregiver participants used the terms 'us' and 'we' when they spoke of diagnostic, treatment and aftercare events, indicating that the experiences had happened to both parties. Patient/caregiver dyads described a team approach to cancer management.

The worst bit was the colonoscopy that you had to get. That was the last follow up after all the procedures we had been through (…later in same interview) The two of you work together and we were lucky [to] both be here at the same time. (Patient 15's husband/caregiver)

At the individual level, participants found that skills learnt during their working lives helped them manage cancer-related work. Taking action was a way to exert control over the illness.

I was the Global trouble shooter of the company, I was on a plane more times than I care to remember, which I don't miss now at all. But a lot of that was problem solving (…) my work ethic was to go on and if there's a problem to be solved you just have to go out and solve it. (Participant 18, male, aged 60–69 years, prostate cancer)

The flexibility to manage personal time (eg, being retired) and financial security protected participants from treatment burden. Conversely, caring responsibilities, coexisting mental health problems or challenging social circumstances increased the risk of treatment burden.

Losing my parents (…) it's an initial domino effect. That causes that, that causes that to cause that (…) I became homeless, that was quite dramatic (…) and then three year ago, 'Oh, you've got prostate cancer' as well to boot and I went, 'Oh, okay, how do you deal with this?' So I was already dealing with things, so like the cancer on top of that, 'Oh, well what else is there?'. (Patient 16, male, aged 60–69 years, prostate cancer)

At the disease level, those with ongoing symptoms or managing complications related to their cancer or comorbidities spent more time managing their health. Several participants reflected on their experiences of supporting friends or loved ones with advanced cancer and believed that treatment workload would be much higher if their disease had been detected at a later stage.

At the health system level, administrative problems were burdensome. Examples included lost appointments, or scans that were not arranged at the expected interval. However, participants with both prostate and colorectal cancers thought that cancer aftercare had been configured to minimise burden for them. Important factors that minimised burden included telephone access to a named specialist nurse or group of nurses; kind, caring and professional staff, close links with local cancer support organisations, timely provision of information and a predetermined, algorithmic treatment and follow-up plan. Participants described being on a '*pathway*' (patient 24), or '*in the hands of the machine*' (patient 24), with '*very [few] rabbit holes that you can jump down*' (patient 22).

### Theme 4: comparisons and interactions between cancer and other comorbidities

Coexisting comorbidities could be more burdensome than cancer. Most participants considered that cancer had been a discrete event which had been cured, whereas other comorbidities such as Parkinson's disease and diabetes required ongoing monitoring, medication adherence and behaviour change. One participant described making significant dietary changes to manage his hypertension but no behavioural changes after a hemicolectomy for screening detected colorectal cancer. Several participants mentioned the 'invisible' nature of cancer, which made it easier to forget.

I mean partly because [psoriasis is] visible and you have to physically treat it yourself, I mean it's putting on the creams and so on and dealing with the… I mean there used to be times when you would bleed (…) whereas with the prostate you don't really have to, well I haven't done anything, I haven't really done anything. (Participant 28, male, aged 70–79 years, prostate cancer)

There was also a sense that cancer care was better resourced and better organised compared with health services for dermatological, respiratory and cardiovascular problems. Two participants suggested that cancer had a high profile in the media and one suggested that cancer was favoured by politicians for financial investment compared with other chronic conditions.

The lasting effects of cancer and its treatments could interact with other comorbidities to increase treatment workload. For example, a participant with Parkinson's disease and tremor found the management of urinary incontinence more challenging.

With the Parkinson's tremor (…) the organising, planning and managing the incontinence (…) is impacting me and can be more frustrating because you have to spend a lot of time on administration (…) In terms of administration, medication (…) how many of these pills have I taken, when do I need to take the next ones, what scenarios could I be in that I need to manage myself [incontinence] and if that happens how do you manage yourself if you're in a group of people and you have to take time out to sort yourself out (…) it's highly frustrating. (Patient 27, male, aged 70–79 years, prostate cancer)

### Theme 5: caring for the caregiver

There was not always a clear distinction between patient and caregiver roles. One caregiver spoke of their own experiences of having cancer and two patient participants were the main caregivers for their wives (who did not participate in this study). These patients experienced the multiplicative burden of managing their own health after cancer while continuing to provide care for their loved one. Both spoke of their role as a caregiver being their main priority.

The weight gain irritates me, and I try to put more exercise in with the indoor cycle, but if I do an indoor cycle for an hour something else has to drop out of my routine in the day, I just don't have the energy to do that, particularly you know, I have the commitment to my wife's care, and I presume that's number one. (Patient 14, male, aged 70–79 years, prostate cancer)

The team approach to cancer management (described in theme 3) affected the caregiver. Some caregivers had taken early retirement or rearranged their working lives to accommodate caring. One caregiver (daughter of patient 19) described driving 40 miles every day to support her parents. Caregivers shared the emotional burden of cancer.

Patient: I think it was the first CT scan, and I came out and I said to [my husband], 'You're looking terrible, you look worried'. He said, 'Well I'm sitting here but I'm in there with you'. It's affecting him just the same, maybe worse at times.

Husband: You're still sitting there waiting (…) your mind's in overtime just wondering how everything's going and hoping everything's okay, so just it affects you both in a different way. (Patient 15, female, aged 60–69 years, colorectal cancer and her husband)

Many men with prostate cancer spoke about the impact that loss of libido and erectile dysfunction had on their relationships. In interviews involving caregiver dyads, sexual dysfunction was acknowledged as '*a source of sadness*' (patient 8), and something which had a continuing effect on both the patient and the caregiver.

We've got the continuing difficulty of my erectile dysfunction (….) We're able to talk openly about it. I don't have the spark anymore, the sensual spark, (…) so that [problem] continues to run for both of us. (Patient 34, male, aged 60–69 years, prostate cancer)

Caregivers spoke of additional difficulties being on the 'side lines' (wife of patient 8) during cancer care, and that, despite playing a key role in treatment decisions and enabling successful recovery, they had limited access to advice or individualised support.

I do think it was frustrating (…) that there wasn't a direct contact that I could've had as the partner, the caregiver because there were some quite concerning moments when he had different reactions (…) it would've been nice to actually speak directly with a medical professional and get some advice. So I did feel a bit lost regarding that (…) and a bit helpless that I couldn't advise and couldn't really help him. (Partner of patient 34)

### Theme 6: the impact of treatment burden

Most of the participants in this study did not perceive themselves to be burdened by cancer aftercare. However, there were three examples of treatment burden influencing cancer treatment and follow-up decisions.

An individual with type 1 diabetes explained that the workload of diabetes had influenced his decision to opt for a radical prostatectomy over more conservative options.

I thought rather than go through [non-surgical treatment for prostate cancer] which can be through monitors and blood tests and maybe more, having to take pills or some kind of medication for your prostate rather than go for the operation, I says, 'Doing insulin five times a day I think is becoming an awful lot and then having any more of a load on top of that', so that was the main reason. (Patient 18, male, aged 60–69 years, prostate cancer)

Another participant with previous primary testicular cancer described '*never being away from the hospital*' when he required hospital treatments for three separate medical conditions. He became aware that his testicular cancer follow-up had not taken place when it should have:

I just couldn't face the thought of going back through, you know, going through all these clinics and everything again, so I left it and never had any follow up for that at all. (Patient 28, male, aged 70–79 years, prostate cancer)

For this participant, perceptions about treatment burden changed over time. During active surveillance for prostate cancer, his other comorbidities were better controlled, and he was older and retired from work. He reflected that, despite spending considerable time travelling to appointments, he did not perceive prostate cancer surveillance to be burdensome.

We stay out in the countryside so every trip into the hospital takes so much longer, it's like if you're going to the hospital for an appointment that's really your day taken up. (…) I don't regard it as an inconvenience, I don't resent it, it doesn't get me annoyed or anything like that, I just think, 'Oh, it's something I've got to do'. Yeah, time passes so quick when you're older, it won't be long before you're back home anyway. (Patient 28, male, aged 70–79 years, prostate cancer)

Two participants (patients 16 and 28) were undergoing active surveillance for prostate cancer and reflected on the invisible nature of the disease. For participant 28, this made it easier to forget about than other comorbidities (see theme 4) but participant 16 experienced psychological burden arising from an invisible disease that could only be monitored through medical tests.

The cancer isn't something that's like a broken leg … It's just something you try to imagine or picture as a fault in the system (…) the only answer to the prostate cancer is biopsies, MRI scans and PSAs…. it's monitoring that the cancer itself if it's on the move

or if it's getting bigger or I don't know if it would ever disappear without treatment. But whether that is contributing to my mental attitude or sometimes the way I think or feel, it's something that's just not very good, not very nice, not nice at all. (Patient 16, male, aged 60–69 years, prostate cancer)

The same individual found prostate biopsies to be '*intrusive*' and '*traumatic*' and started to disengage with monitoring.

I haven't seen a doctor, I don't want to see a doctor. (…) it was traumatic enough to say that I'm just sick of this idea of giving biopsies, seeing doctors. And I even chickened out, I don't know if that's the word, I even missed giving a PSA on the three monthly after May. (Patient 16, male, aged 60–69 years, prostate cancer)

## DISCUSSION
### Main findings
The term 'burden' did not resonate strongly with most cancer survivors in this study, despite descriptions of time-intensive treatment and follow-up regimens and the ongoing management of problematic symptoms such as fatigue and incontinence. The notion of burden was incongruent with the gratitude that cancer survivors expressed for curative treatment. Time invested in cancer management would ultimately lead to survival and more time to spend on meaningful activities.

Cancer was framed as a discrete episode—something invisible that had been cut out or contained and that should be consigned to the past. In this sample, cancer was not perceived to require the lifelong adherence to self-monitoring and lifestyle changes that were emphasised in conditions such as diabetes or hypertension. Cancer was not considered to be a chronic disease and none of the participants mentioned work to prevent recurrence.

There were potentially modifiable factors that could make treatment burden more or less likely, including social support, financial stability and health system configuration. In this sample, the most important factors were interacting comorbidities, which increased burden, and the presence of a caregiver, which reduced treatment burden. Caregivers shared in all aspects of the work of cancer survivorship care, and in the emotional and practical consequences of this work such as experiencing worry and disruption to working lives.

Despite few patients perceiving cancer management to be burdensome, there were three examples in which treatment burden had influenced cancer treatment decisions or disengagement with follow-up. Interestingly, individuals with similar treatment regimens had very different perceptions of burden, and perceptions of treatment burden could change over time. Disengagement with cancer monitoring is a cause for concern and is a feasible

mechanism through which treatment burden could negatively affect cancer outcomes.

### Comparison with existing literature
Treatment burden has mainly been researched in multimorbidity[37] and in cardiovascular disease,[38] diabetes[39] and stroke.[11] Patients are known to be burdened by 'fragmented' medical care, poor communication/lack of empathy from health professionals and inadequate information provision.[40] These problems were less evident in our sample, and it was suggested by participants that cancer services are prioritised over other chronic diseases because of their prominence in the media and public eye.

A recent scoping review found that financial burden, time/travel burden and medication burden were the most prominent dimensions of treatment burden in older individuals with cancer.[16] Several studies have focused on individual factors which can contribute to treatment burden and diminished quality of life, such as financial toxicity, time spent on cancer treatments and the burden of adhering to medications.[15 41 42] In our study, treatment burden was considered as a multidimensional concept. Financial burden and time/travel burden were not significant problems in our participants, despite almost half the sample living in rural areas. This emphasises that burden is a subjective perception that is influenced by multiple interacting factors.

We identified patient, disease and health system-level factors that can increase or decrease treatment burden. Many of these fit well with established theory, which suggests a 'cumulative complexity' of healthcare work.[43] Burden occurs when patient workload and demand exceeds patient capacity to undertake this work.[7 43] A prominent difference in our sample of individuals with good prognosis cancers compared with other chronic diseases like stroke or heart failure was that the prospect of cure and extending life enhanced patients' psychological and motivational capacity to engage with health-related work. This diminished the perception of burden.

An interesting question arising from this work is whether all cancers should be considered as chronic or long-term conditions. Long-term conditions are '*conditions for which there is currently no cure, and which are managed with drugs and other treatment*'.[44] Individuals in this study had been treated with curative intent. However, all participants were at risk of recurrence, many had lasting effects of treatment and all were at slightly higher risk of second primary cancers.[45 46] A challenge for survivorship care is how to introduce the nuance of long-term health management against the binary messages of 'curable' or 'incurable' that are presented during active treatment.

In this sample, opportunities to promote exercise, healthy diet and weight management were being missed but patients had capacity to take on this work.

### Strengths and limitations
This study is one of the first to specifically investigate burden of treatment and its impact in survivors of good

prognosis cancers and their caregivers. It adds granular, mechanistic details to previous quantitative observations that comorbidities and social support can influence patient ratings of treatment burden.[20 22] The selected cancers encompassed a wide variety of treatment modalities, different follow-up regimens and a spectrum of illness burdens. Over 25 hours of rich audio-recorded data were generated from in-depth interviews. The involvement of four authors from different backgrounds enriched the analysis with different theoretical and methodological perspectives. Results were fed back to participants to ensure they reflected participant experiences, and that important findings were not omitted.

There are important limitations to note. As prostate cancer is a male cancer, men are over-represented in this sample. Gender is an important antecedent of treatment burden,[17] but gender issues were not specifically probed during interviews. Men gave detailed and open accounts of gender-specific problems relating to hormonal treatment and prostate surgery, such as erectile dysfunction, loss of libido and urinary symptoms. However, it is possible that broader issues relating to gender and treatment burden have been missed. Some research has suggested that women may experience more treatment burden than men,[22 47–49] but the contribution of gender and identity to treatment burden is under-researched.[50] The addition of a female cancer in this study might have highlighted important gender-related differences in treatment burden experiences.

Only six caregivers were recruited, five of whom were female. Despite this, caregivers contributed meaningful insights into their role and experiences of treatment burden, and it became clear that patient and caregiver roles were not mutually exclusive. In future research, it would be important to target caregivers through specific channels of recruitment. Interviewing caregivers on their own could also highlight aspects of their experience that they might be less likely to discuss with their loved one.

All participants were from a single geographical area, attending an academic teaching hospital and most were from areas of low socioeconomic deprivation with relatively low levels of multimorbidity. Multimorbidity and socioeconomic deprivation are significant risk factors for treatment burden. This study may have underestimated the impact of treatment burden on cancer survivors. There is a paradox that those who are significantly burdened by their treatment may have less capacity to participate in research, and it is important to consider mechanisms of incentivising and including these individuals in future research.

### Implications for cancer survivors, practice and future research

Cancer survivors can be reassured that treatment burden tends to decrease over time after active treatment, and that most of the individuals in this study were not burdened by health-related workload. Nevertheless, there were indications that treatment burden could drive inequities in cancer outcomes, particularly in individuals with limited social support and concurrent comorbidities. Clinicians should carefully consider what they are asking patients and their caregivers to do, and whether they have the capacity to undertake this work.

Future research might focus on innovative ways to provide accessible, structured, holistic care to multimorbid cancer survivors. There may be an increasing role for 'specialist generalists'.[51] Future research should examine the relationship between treatment burden and specific, measurable outcomes after cancer such as survival, recurrence and quality of life. Interventions might target those most at risk of treatment burden to improve their outcomes after cancer.

## CONCLUSIONS

There is a continuum between positive perceptions of health-related work and burden in cancer survivors. A cancer diagnosis serves as a strong motivator to engage in health management, and perceptions about burden can be shifted by individual, disease and health system factors. Treatment burden can affect engagement with and decisions about care and is an important consideration in cancer survivorship care.

**Acknowledgements** The authors thank the participants who volunteered their time to give detailed accounts of their experiences. We also thank the National Health Service Research Scotland Primary Care Network, who facilitated patient recruitment via general practices. We thank the general practices who participated.

**Contributors** Conceptualisation: RA. Data collection: RA, LD. Data analysis: RA, LD, SJM, LL. Writing—original draft: RA. Revision of manuscript critically: RA, LD, SJM, LL. RA is the guarantor and accepts full responsibility for the work and the conduct of the study. RA had full access to the data and controlled the decision to publish.

**Funding** This work was funded by a Scottish Government Chief Scientist Office Senior Clinical Academic Fellowship awarded to RA (reference: SCAF/18/02).

**Disclaimer** The funder did not play a role in data collection, interpretation or reporting.

**Competing interests** None declared.

**Patient and public involvement** Patients and/or the public were involved in the design, or conduct, or reporting, or dissemination plans of this research. Refer to the Methods section for further details.

**Patient consent for publication** Not applicable.

**Ethics approval** This study involves human participants and was approved by the North of Scotland Research Ethics Committee (reference: 19/NS/0158).

**Provenance and peer review** Not commissioned; externally peer reviewed.

**Data availability statement** No data are available. Participants of this study did not agree for their interview transcripts to be shared publicly, so supporting data are not available.

**ORCID iDs**
Rosalind Adam http://orcid.org/0000-0003-3082-6578
Lisa Duncan http://orcid.org/0000-0002-7768-3909

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
