## [Reviewer comments · BMJ Open]

ARTICLE DETAILS

TITLE (PROVISIONAL)	Treatment burden in survivors of prostate and colorectal cancer: a qualitative interview study
AUTHORS	Adam, Rosalind; Duncan, Lisa; Maclennan, Sara; Locock, Louise

VERSION 1 – REVIEW

REVIEWER	Thong, Melissa German Cancer Research Center
REVIEW RETURNED	05-Dec-2022

GENERAL COMMENTS	This is a well-written paper that aimed to shed light on the treatment burden of survivors of prostate and colorectal cancer through semi-structured interviews. Some thoughts arising from the reading of the manuscript: 1. Caregivers were also interviewed for this study, providing a potential valuable source of information. However, it is a pity that only 6 caregivers (of whom 5 are female) participated. This sample may not adequately cover the concerns/treatment burden from the perspective of the caregivers. Since it was stated as one of the objectives of this study (page 6), the small sample and mainly female caregivers could be a potential limitation that is not mentioned.
---

REVIEWER	Sav, Adem Queensland University of Technology, Faculty of Health
REVIEW RETURNED	06-Dec-2022

GENERAL COMMENTS	Thank you for the opportunity to review this paper. I think it focuses on a very important area of research and is well written - well done! It does need a bit more work though, particularly in the findings/discussion. I have provided comments throughout the paper which may be useful Thank you - The reviewer provided a marked copy with additional comments. Please contact the publisher for full details.
---

VERSION 1 – AUTHOR RESPONSE

REVIEWER ONE

Comment: This is a well-written paper that aimed to shed light on the treatment burden of survivors of prostate and colorectal cancer through semi-structured interviews.

Response: Thank you.

Comment: Caregivers were also interviewed for this study, providing a potential valuable source of information. However, it is a pity that only 6 caregivers (of whom 5 are female) participated. This sample may not adequately cover the concerns/treatment burden from the perspective of the caregivers. Since it was stated as one of the objectives of this study (page 6), the small sample and mainly female caregivers could be a potential limitation that is not mentioned.

Response: We agree that this is a limitation. It was also the case that we had not adequately represented the caregiver views and opinions in the original draft of this manuscript. We had rich caregiver data that were not presented. We have returned to the original data and have added a theme: “caring for the caregiver” as follows:

“Theme 5: caring for the caregiver

There was not always a clear distinction between patient and caregiver roles. One caregiver spoke of their own experiences of having cancer and two patient participants were the main caregivers for their wives (who did not participate in this study). These patients experienced the multiplicative burden of managing their own health after cancer whilst continuing to provide care for their loved one. Both spoke of their role as a caregiver being their main priority.

“The weight gain irritates me, and I try to put more exercise in with the indoor cycle, but if I do an indoor cycle for an hour something else has to drop out of my routine in the day, I just don’t have the energy to do that, particularly you know, I have the commitment to my wife’s care, and I presume that’s number one.” Patient 14, male aged 70-79 years, prostate cancer.

The team approach to cancer management (described in theme 3) affected the caregiver. Some caregivers had taken early retirement or rearranged their working lives to accommodate caring. One caregiver (daughter of patient 19) described driving 40 miles every day to support her parents. Caregivers shared the emotional burden of cancer.

Patient: I think it was the first CT scan, and I came out and I said to [my husband], “You’re looking terrible, you look worried”. He said, “Well I’m sitting here but I’m in there with you”. It’s affecting him just the same, maybe worse at times”.

Husband: You’re still sitting there waiting (...) your mind’s in overtime just wondering how everything’s going and hoping everything’s okay, so just it affects you both in a different way. Patient 15, female aged 60-69 years, colorectal cancer and her husband.

Many men with prostate cancer spoke about the impact that loss of libido and erectile dysfunction had on their relationships. In interviews involving caregiver dyads, sexual dysfunction was acknowledged as “a source of sadness” (patient 8), and something which had a continuing effect on both the patient and the caregiver.

“We’ve got the continuing difficulty of my erectile dysfunction (...) We’re able to talk openly about it. I don’t have the spark anymore, the sensual spark, (...) so that [problem] continues to run for both of us.” Patient 34, male aged 60-69 years, prostate cancer.

Caregivers spoke of additional difficulties being on the “side lines” (wife of patient 8) during cancer care, and that, despite playing a key role in treatment decisions and enabling successful recovery, they had limited access to advice or individualised support.

“I do think it was frustrating (...) that there wasn’t a direct contact that I could’ve had as the partner, the caregiver because there were some quite concerning moments when he had different reactions (...) it would’ve been nice to actually speak directly with a medical professional and get some advice. So I did feel a bit lost regarding that (...) and a bit helpless that I couldn’t advise and couldn’t really help him”. Partner of patient 34.”

We have also added the following statement to the limitations section of the discussion:

“Only six caregivers were recruited, five of whom were female. Despite this, caregivers contributed meaningful insights into their role and experiences of treatment burden, and it became clear that patient and caregiver roles were not mutually exclusive. In future research it would be important to target caregivers through specific channels of recruitment. Interviewing caregivers on their own could also highlight aspects of their experience that they might be less likely to discuss with their loved one.”

REVIEWER TWO

Comment: Thank you for the opportunity to review this paper. I think it focuses on a very important area of research and is well written - well done! It does need a bit more work though, particularly in the findings/discussion. I have provided comments throughout the paper which may be useful Thank you

Response: Thank you. We have considerably re-worked the paper, returning to the original data and adding details in all sections but particularly in the results and discussion sections.

Comment (Introduction): I think you need a bit more information on describing and defining treatment burden. How is treatment burden different from illness burden? What are the major themes/concepts of treatment burden? Etc.

Response: We have added the following text to the introduction section of the manuscript.

“Treatment burden is the workload of healthcare and the impact that this work has on the individual [7]. Treatment burden is of increasing importance [8] due to ageing populations, a rising prevalence of multimorbidity [9, 10], and increased pressure on healthcare systems. Healthcare workload can encompass a wide range of tasks, including “sense-making” work [11], monitoring/managing symptoms, managing medicines, navigating the healthcare system, and changing health-related behaviours [8, 11].

Treatment burden and illness/disease burden are closely linked but conceptually different. Illness burden describes the impact of an illness on an individual, such as morbidity and mortality [12]. The actions taken to manage health and their consequences can lead to treatment burden [13–15].

Sav et al noted six key domains of treatment burden, encompassing “financial, medication, administrative, time/travel, lifestyle, and healthcare” dimensions, and “antecedents” which can influence the severity of treatment burden, such as age, gender, treatment characteristics, and disease type [16, 17]. Having good social support or a caregiver can lower treatment burden for patients [17], but caregivers can also become burdened [18, 19]. The impact of treatment burden on informal caregivers is under-researched [18].”

Comment (Introduction, line 49): What about different types of cancers?

Response: We have changed this sentence to read: “It is unclear how treatment burden might be perceived by survivors of different types of cancer in whom the prognosis is more favourable than lung cancer.”

Comment (regarding prostate and colorectal cancers being ideal exemplar cancers): I would not completely agree. Given that they are more focused on men, there will be issues of masculinity that come into play, which have relevance for treatment burden - this needs to be acknowledged.

Response: We have removed the statement that they are ideal exemplar cancers and instead have explained in more detail why they were chosen for this study:

“Prostate and colorectal cancers were chosen to explore treatment burden in this study because they encompass a wide range of treatment modalities (surgery, radiotherapy, chemotherapy), lasting sequelae (e.g., fatigue, persistent pain, incontinence, sexual problems, and stoma management), and follow-up activities. Individuals play a key role in improving their own prognosis by self-monitoring for symptoms and attending for scans and blood tests to detect recurrence, and by adhering to diet and exercise recommendations [4, 27], and may therefore be at risk of treatment burden. Informal caregivers are key supporters of these activities [28].”

We have added a reflection on gender to the discussion section (see later responses).

Comment: I think you need to introduce caregivers earlier and justify why they need to be investigated in this study

Response: We agree and have added detail to several paragraphs of the introduction to justify the inclusion of caregivers, including what is already known about the role of caregivers, and the fact that the role of caregivers is under-researched. Additional references have been added throughout the introduction section.

Comment (Introduction, final paragraph): So what is the relevance of the study and potential implications?

Response: We have added the following text to the final paragraph of the introduction section:

“It is held that individuals who become overburdened by the workload of health care disengage from self-management activities, leading to poorer outcomes [21–23]. Treatment burden could be an important mediator of poorer outcomes in cancer survivors and patients/caregivers are best-placed to give insights into mechanisms of treatment burden and aspects that are potentially modifiable.”

Comment: NHS – used first time

Response: We have added “National Health Service” the first time this abbreviation is used.

Comment (methods): A little bit more info on the setting would be good.

Response: We have added the following information:

“The NHS is a publicly funded healthcare system which is free at the point of delivery. In Grampian, cancer care is centred around a university teaching hospital in Aberdeen with academic links, and care pathways that are integrated with local cancer charities [29]. Grampian had a 2011 census population of 569,061, and around one third of the population live rurally [30].”

Reference 29 contains a lot of additional detail on the location and about urology cancer services in Grampian.

Comment (methods): five years is a long time when you ask people to remember things

Reference: A cut-off of five years after diagnosis was chosen because patients are all still actively undergoing secondary care follow up for their cancer during this period, so although it may have been up to five years since diagnosis, individuals were still attending for regular scans, blood tests, and appointments relating to the cancer. The “survivorship” period after completion of active treatment is under-researched but potentially important in terms of treatment burden. It is also important to note that participants were all at different points in their trajectory of follow-up and aftercare – some had finished treatment very recently whereas others were in their fifth year after diagnosis. This gave us rich data about the whole follow-up period.

Comment (methods): do you think those active surveillance only will have different experiences?

Response: we included active surveillance because, by definition, these individuals have relatively low risk cancers with excellent prognoses, but still require many of the same follow up procedures (e.g. PSA blood testing, regular hospital appointments) as those men who have had surgery, radiotherapy, or hormone treatment. We had two men in our sample who were undergoing active surveillance and we have included some additional quotations from these men in the results section, particularly highlighting the “invisible” nature of the disease, and the psychological burden of active surveillance (see quotations below in subsequent comments).

Comment (methods): what about carers? Who were they?

Response: We invited patients to identify a caregiver and did not provide them with a specific definition of the term. The following sentence has been added to the methods section:

“Eligible patients were invited to nominate a caregiver to participate in the interview. Separate invitation letters and information sheets were included for caregivers in packs sent to patients.”

Comment (methods): would be nice to attach this also (interview schedule)

Response: We have added the interview guide as a supporting file.

Comment (methods): Why these models? (NPT/HAPA)

Response: We have added an explanation for why these models were chosen, as follows:

“NPT has been used to understand how individuals embed new practices within their daily life and has been a useful model through which to explore treatment burden after stroke [11]. The HAPA describes both how individuals become motivated to engage with healthcare or self-management work, and how individuals then translate this motivation into engagement with and maintenance of self-management practices over time. The HAPA integrates and extends previous behavioural theories by including a range of important constructs such as self-efficacy and intention, which can predict and explain human behaviours [33].”

Comment (methods): How was this (interview schedule) adjusted for caregivers?

Response: Experienced interviewers used the guide flexibly, using the same headings/questions but tailoring the wording for caregivers and ensuring that caregivers were given the full opportunity to express their views and opinions. We have added the following sentence to the paper:

“ Participants and caregivers were asked about the same topics and interviewers adapted the questions during the interview to ensure that caregiver perceptions were fully captured.”

Comment: Reference NVivo

Response: The reference has been added.

Comment (methods): What did you do specifically to increase the trustworthiness of the data?

Response: Interviews were transcribed verbatim by a professional transcription company and transcripts were checked in full for accuracy, mainly by the lead author. The robustness of the analysis was enhanced by inclusion of four authors who met regularly to discuss themes, and line-by-line coding of transcripts by the first two authors. Participants were also given the chance to provide feedback on our results.

Comment (methods): What did they (participants who provided feedback) say?

Response: We have added the following sentence:

“Three participants replied and provided general updates on their progress. None suggested any changes or additions to comments on the results.”

Comment (results): so your sample was skewed towards males - need to acknowledge this as it will have an impact on treatment burden experiences. My experience is that men are less likely to admit to tx burden than women.

Response: we have added the following text to the limitations section of the discussion section:

“There are important main limitations to note. As prostate cancer is a male cancer, men are over-represented in this sample. Gender is an important antecedent of treatment burden, [17] but gender issues were not specifically probed during interviews. Men gave detailed and open accounts of gender-specific problems relating to hormonal treatment and prostate surgery, such as erectile dysfunction, loss of libido and urinary symptoms. However, it is possible that broader issues relating to gender and treatment burden have been missed. Some research has suggested that women may experience more treatment burden than men [22, 47-49], but the contribution of gender and identity to treatment burden is under-researched [50]. The addition of a female cancer in this study might have highlighted important gender-related differences in treatment burden experiences.”

Comment (results): sequelae of cancer (?)

Results: we have provided examples of the sequelae of cancer in the introduction as follows:

“Prostate and colorectal cancers were chosen to explore treatment burden in this study because they encompass a wide range of treatment modalities (surgery, radiotherapy, chemotherapy), lasting sequelae (e.g., fatigue, persistent pain, incontinence, sexual problems, and stoma management), and follow-up activities.”

Comment (results): Any example quotes (regarding active psychological approaches to adjustment after cancer and work required to return to near normal)?

Response: We have added the following quote:

“*I do feel that I totally lost two years of my life, because I think it took that time to get through the operation, get back to feeling normal (...) I think everything had changed about my life, in a way it had stopped and I've started a new life now (...) you think things will never be normal again, but you find a new normal.*” Participant 15, female aged 60-69 years, colorectal cancer.”

Comment (results, theme 3): I don't really think this theme is about empowerment.

Response: We revisited the original data and would agree that “empowerment” is not a key concept. We were trying to convey that there is a spectrum of positive to negative perceptions about cancer management work. Some participants felt more in control of their disease by undertaking follow up and self management work. We have renamed the theme to: “*factors modifying the experience of treatment burden*”. We have updated Figure 1 accordingly.

Comment (results, disease-level factors influencing treatment burden): what about comorbidity?

Response: We agree that comorbidity is an important addition at the disease-level and have included “comorbidity” in this sentence. We also have a dedicated theme on comorbidity (theme 4).

Comment (Results, theme 3): Just wondering which participant said this (being on a “pathway”, or “in the hands of the machine”, with “very [few] rabbit holes that you can jump down”).

Response: We have added participant IDs to these quotations.

Comment (results, theme 6, the impact of treatment burden): I wonder if there were any gender differences?

Response: All the participants who described treatment burden having an influence on cancer treatment and follow up decisions were men with prostate cancer. However, as prostate cancer is a disease of men, our sample is skewed towards male participants. We have highlighted this in the “limitations” section. In terms of the types of impact, one participant was burdened by injecting insulin five times daily, one was burdened by frequent hospital attendances for comorbidities, and one was burdened by “intrusive” and “traumatic” biopsies. There was no signal that these were gender-specific problems but we acknowledge that we are not able to fully explore the role of gender in this study. It would be interesting to conduct a study that focuses on gender and identity issues relating to treatment burden after cancer and to include, for example, gynaecological and breast cancers as well as male-specific cancers.

Comment (Results, theme 6) “He became aware that his testicular cancer follow-up had not taken place when it should have”): Make sure grammar is consistent

Response: Thank you. We have proofread the document in full now and have checked for grammatical and typographical errors. We are happy that these have been addressed.

Comment (Results, theme 6): Given that your sample is quite large for a qualitative study, I wonder if your findings can be a bit more in-depth and offer more insight.

Response: We have added additional details and examples of the impact of treatment burden in theme 6 as follows:

“Another participant with previous primary testicular cancer described “*never being away from the hospital*” when he required hospital treatments for three separate medical conditions. He became aware that his testicular cancer follow-up had not taken place when it should have:

“I just couldn’t face the thought of going back through, you know, going through all these clinics and everything again, so I left it and never had any follow up for that at all.” Patient 28, male aged 70-79 years, prostate cancer.

For this participant, perceptions about treatment burden changed over time. During active surveillance for prostate cancer, his other comorbidities were better controlled, and he was older and retired from work. He reflected that, despite spending considerable time travelling to appointments, he did not perceive prostate cancer surveillance to be burdensome.

“We stay out in the countryside so every trip into the hospital takes so much longer, it’s like if you’re going to the hospital for an appointment that’s really your day taken up. (...) I don’t regard it as an inconvenience, I don’t resent it, it doesn’t get me annoyed or anything like that, I just think, “Oh, it’s something I’ve got to do”. Yeah, time passes so quick when you’re older, it won’t be long before you’re back home anyway.” Patient 28, male aged 70-79 years, prostate cancer.

Two participants (patients 16 and 28) were undergoing active surveillance for prostate cancer and reflected on the invisible nature of the disease. For participant 28, this made it easier to forget about than other comorbidities (see theme four) but participant 16 experienced psychological burden arising from an invisible disease that could only be monitored through medical tests.

The cancer isn’t something that’s like a broken leg ... It’s just something you try to imagine or picture as a fault in the system (...) the only answer to the prostate cancer is biopsies, MRI scans and PSAs.... it’s monitoring that the cancer itself if it’s on the move or if it’s getting bigger or I don’t know if it would ever disappear without treatment. But whether that is contributing to my mental attitude or sometimes the way I think or feel, it’s something that’s just not very good, not very nice, not nice at all. Patient 16, male aged 60-69 years, prostate cancer.

The same individual found prostate biopsies to be “*intrusive*” and “*traumatic*” and started to disengage with monitoring.

I haven't seen a doctor, I don't want to see a doctor. (...) it was traumatic enough to say that I'm just sick of this idea of giving biopsies, seeing doctors. And I even chickened out, I don't know if that's the word, I even missed giving a PSA on the three monthly after May” Patient 16, male aged 60-69 years, prostate cancer.

We have also added significant details about the role of the caregiver (see above).

Comment (discussion – cancer as a discrete episode – something invisible that had been cut out or contained and that should be consigned to the past): I don't think this really emerged from your findings and needs to be made clearer (in the findings)

Response: In theme four we have the following text: “Most participants considered that cancer had been a discrete event which had been cured, whereas other comorbidities such as Parkinson's disease and diabetes required ongoing monitoring, medication adherence, and behaviour change. One participant described making significant dietary changes to manage his hypertension but no behavioural changes after a hemicolectomy for screening detected colorectal cancer. Several participants mentioned the “invisible” nature of cancer, which made it easier to forget.” We also speak about returning to “normality” in theme one. In theme 6, impact of treatment burden, we have reflected more on the “invisible” nature of cancer (see above).

Comment (discussion, comparison with existing literature): there has been significant research around tx burden and cancer now also - please look and cite and explain how your study adds to or extends this literature. I don't think your findings really reflect (the cumulative complexity model). Despite the workload involved with cancer it appears (as you say) it was not seen burdensome because of the appreciating nature of treatment to prolong life - I think this is a major difference with other ongoing non-curable chronic conditions such as diabetes. - this needs to be explained more explicitly in your discussion.

Response: We have reworked the “comparison with existing literature” section to include references to research around treatment burden in cancer, how our study adds to and extends this literature, and to explicitly state the difference between cancer and other non-curable conditions, as follows:

“A recent scoping review found that financial burden, time/travel burden and medication burden were the most prominent dimensions of treatment burden in older individuals with cancer [16]. Several studies have focused on individual factors which can contribute to treatment burden and diminished quality of life, such as financial toxicity, time spent on cancer treatments, and the burden of adhering to medications [15, 41, 42]. In our study, treatment burden was considered as a multidimensional concept. Financial and time/travel burden were not significant problems in our participants, despite almost half the sample living in rural areas. This emphasises that burden is a subjective perception that is influenced by multiple interacting factors.

We identified patient-, disease-, and health system level factors that can increase or decrease treatment burden. Many of these fit well with established theory, which suggests a “cumulative complexity” of health care work [43]. Burden occurs when patient workload and demand exceeds patient capacity to undertake this work [7, 43]. A prominent difference in our sample of individuals with good prognosis cancers compared to other chronic diseases like stroke or heart failure was that the prospect of cure and extending life enhanced patients psychological and motivational capacity to engage with health-related work. This diminished the perception of burden.”

Comment (discussion): it was suggested by participants that cancer services are prioritised over other chronic diseases: why do you think this was the case?

Response: We have added the following:

“ it was suggested by participants that cancer services are prioritised over other chronic diseases because of their prominence in the media and public eye.”

Comment (discussion): I think the gender issue in relation to tx burden is missing in your paper and needs to be addressed.

Response: please see detailed response above.

Comment (discussion): I think the carers in your study seem to be a lost voice. Even though there were only limited number of carers, I don't think they got the attention they deserved in the paper.

Response: please see our response to reviewer one.

Comment (strengths and limitations): “this study adds new insights into burden of treatment in cancer survivors”: how? Please be specific.

Response: We have added the following:

“This study is one of the first to specifically investigate burden of treatment and its impact in survivors of good prognosis cancers and their caregivers. It adds granular, mechanistic details to previous quantitative observations that comorbidities and social support can influence patient-ratings of treatment burden [20, 22].”

Comment (discussion – strengths and limitations, multimorbidity and socioeconomic deprivation are significant risk factors for treatment burden.): what about those from rural and remote locations who need to travel for cancer treatment?

Response: we were able to include a good mix of urban and rural patients, including individuals from remote rural locations. We did not find travelling time to be a significant burden to rural patients. We have added a reflection on this in the “comparison with existing literature” section (see above).

Comment (discussion, implications for practice and future research): what about implications for cancer survivors or participants?

Response: We have added the following:

Implications for cancer survivors, practice, and future research “

Cancer survivors can be reassured that treatment burden tends to decrease over time after active treatment, and that most of the individuals in this study were not burdened by health-related workload. Nevertheless, there were indications that treatment burden could drive inequities in cancer outcomes, particularly in individuals with limited social support and concurrent comorbidities. Clinicians should carefully consider what they are asking patients and their caregivers to do, and whether they have the capacity to undertake this work.

VERSION 2 – REVIEW

REVIEWER	Thong, Melissa German Cancer Research Center
REVIEW RETURNED	08-Feb-2023

GENERAL COMMENTS	Thank you for the revision.
-----------------------------

REVIEWER	Sav, Adem Queensland University of Technology, Faculty of Health
REVIEW RETURNED	06-Feb-2023

GENERAL COMMENTS	Well done for comprehensively addressing the comments and improving the paper.
--